# Trends in urinary tract infection hospitalization in older adults in Spain from 2000-2015

Jesús Redondo-Sánchez[1,2☉], Isabel del Cura-González[2,3,4☉], Laura Díez-Izquierdo[5], Ricardo Rodríguez-Barrientos[3,4,6]*, Francisco Rodríguez-Cabrera[7], Elena Polentinos-Castro[3,4], Miguel López-Miguel[8], Lucas Marina-Ono[9], Laura Llamosas-Falcón[10], Ángel Gil-de Miguel[2]

**1** Ramon y Cajal Health Care Center, Alcorcón, Primary Care Management, Madrid Health Service, Madrid, Spain, **2** Department of Medical Specialties and Public Health, University Rey Juan Carlos, Alcorcón, Madrid, Spain, **3** Research Unit, Primary Care Management, Madrid Health Service, Madrid, Spain, **4** Health Services Research on Chronic Patients Network (REDISSEC) ISCIII, Madrid, Spain, **5** Infanta Sofía University Hospital, San Sebastián de los Reyes, Madrid, Spain, **6** Biosanitary Research and Innovation Foundation of Primary Care (FIIBAP), Madrid, Spain, **7** National School of Health, Instituto de Salud Carlos III, Madrid, Spain, **8** Ciudades Health Care Center, Getafe, South Family and Community Care Teaching Unit, Madrid, Spain, **9** Getafe University Hospital, Madrid, Spain, **10** Preventive Medicine and Public Health, 12 de Octubre University Hospital, Madrid, Spain

☉ These authors contributed equally to this work.
* ricardo.rodriguez@salud.madrid.org

**Data Availability Statement:** Ethics Committee of the Hospital Universitario Fundación Alcorcón has approved this research, including any potential data

## Abstract

### Objective

To analyze trends in urinary tract infection hospitalization (cystitis, pyelonephritis, prostatitis and non-specified UTI) among patients over 65 years in Spain from 2000–2015.

### Methods

We conducted a retrospective observational study using the Spanish Hospitalization Minimum Data Set (CMBD), with codifications by the International Classification of Diseases (ICD-9). We collected data on sex, age, type of discharge, main diagnosis, comorbid diagnosis, length of stay, and global cost. All the hospitalizations were grouped by age into three categories: 65–74 years old, 75–84 years old, and 85 years old and above. In the descriptive statistical analysis, crude rates were defined as hospitalizations per 1,000 inhabitants aged ≥65. To identify trends over time, we performed a Joinpoint regression.

### Results

From 2000–2015, we found 387,010 hospitalizations coded as UTIs (54,427 pyelonephritis, 15,869 prostatitis, 2643 cystitis and 314,071 non-specified UTI). The crude rate of hospitalization for UTIs between 2000 and 2015 ranged from 2.09 in 2000 to 4.33 in 2015 Rates of hospitalization were higher in men than in women, except with pyelonephritis. By age group, higher rates were observed in patients aged 85 years or older, barring prostatitis-related hospitalizations. Joinpoint analyses showed an average annual percentage increase

sharing. The CMBD data belong to the Ministry of Health and are partially accessible to the public. For data series on hospitalization, like the data presented in this article, specific selections of anonymized microdata from CMBD records can be requested from the Ministry. The application form is at https://www.mscbs.gob.es/estadEstudios/ estadisticas/estadisticas/estMinisterio/ SolicitudCMBD.htm.

**Funding:** This study has been funded by Instituto de Salud Carlos III through the project "PI19/ 01700", as part of the Plan Estatal de I+D+I 2017-2020 co-funded by European Regional Development Fund (ERDF) "A way of shaping Europe". In addition, the principal investigator JRS received support to increase his research activities and to publish this manuscript from the 2020 funding program of the Fundación de Investigación e Innovación Biosanitaria en Atención Primaria (FIIBAP), Community of Madrid.

**Competing interests:** The authors have declared that no competing interests exist.

(AAPC) in incidence rates of 4.9% (95% CI 3.2;6.1) in UTI hospitalizations. We observed two joinpoints, in 2010 and 2013, that found trends of 5.5% between 2000 and 2010 (95% CI 4.7;6.4), 1.5% between 2010 and 2013 (95% CI -6.0;9.6) and 6.8% between 2013 and 2015 (95% CI -0.3;14.4).

## Conclusions

The urinary infection-related hospitalization rate in Spain doubled during the period 2000–2015. The highest hospitalization rates occurred in men, in the ≥85 years old age group, and in non-specified UTIs. There were increases in all types of urinary tract infection, with non-specified UTIs having the greatest growth. Understanding these changing trends can be useful for health planning.

## Introduction

Urinary tract infections (UTIs) are one of the most frequent community-acquired bacterial infections [1, 2]. They account for between 10 and 15% of hospitalizations for infectious causes [3, 4] and up to 6% of infection-related deaths [4].

In recent years, hospital admissions due to UTIs have increased by up to 50% [5–7]. These hospitalizations have a high economic impact, primarily associated with the length of the hospital stay [8]. The risk factors associated with community-acquired UTIs that require hospitalization include male sex [9], high prevalence of comorbidities such as dementia, stroke or urinary incontinence [10] instrumentation of the urinary tract, polypharmacy and highly resistant pathogens associated with treatment failure [11, 12].

As well as these conditioning factors, age is a risk factor for hospitalization [13] and the best predictor of admission-related mortality [14, 15]. Previous community-level population studies have revealed an increasing trend of UTIs with age [16], reporting a higher risk of bacteremia and death by all causes in male patients and patients above 85 years, especially in depressed areas [17].

There are no studies on hospitalization incidence due to UTIs in Spain. From an epidemiological standpoint, studying these data requires not only a global approach but also stratification by sex and age categories, as these entail distinctive features that impact the outcome [6]. It is important to understand the data on UTI incidence at the hospital and community levels when planning and implementing policies for antibiotic use [18].

Considering the aging population, our primary hypothesis was that there would be an increase in the number of hospital admissions in total UTIs and in each type of UTI over the study period. The objective of this study was to analyze trends in hospitalizations related to urinary infections in patients above 65 years of age, over a period of 16 years, according to their diagnoses of cystitis, pyelonephritis, acute prostatitis, and non-specified UTIs.

## Methods

We conducted a retrospective observational study using the Spanish Hospitalization Minimum Data Set (CMBD). The Reporting of Studies Conducted using Observational Routinely-collected Health Data (RECORD checklist) is available as supporting information; see S1 File.

## Participants and data source

Our study includes hospitalizations from 2000 to 2015, as during this time span, codifications remained homogeneous nationally, following the International Classification of Diseases, ninth version (ICD-9). Currently, approximately 99% of the population in Spain has National Health Service (SNHS) public coverage. The CMBD is a registry for all hospitals in Spain that contains data for up to 97.7% of discharges from public hospitals during all periods, and from 2005, the CMBD had gradual coverage from private hospitals, reaching a total coverage from all hospitals of 92% in 2014. The CMBD defines hospitalization as admission to the emergency department for over 24 hours and/or scheduled admission. Admissions that occur within 30 days of discharge are considered to be readmissions. The CMBD has proven its usefulness in previous epidemiological studies [19, 20].

From the CMBD database, we selected hospitalizations in which their main diagnosis was urinary tract infection. We considered UTI as the diagnosis codes for pyelonephritis (590.10-11-3-80-81-9), acute (601.0) and chronic (601.1) prostatitis, cystitis (595.0-89-9), and non-specified UTI (599.0). Among hospitalizations with the main diagnosis of non-specified UTI, those who had a secondary diagnosis of pyelonephritis, acute or chronic prostatitis, cystitis, or bacteriuria were reclassified as such to maximize the chances of identifying specific diagnosis of UTI.

For each hospitalization, we collected data on patients' demographics (sex, age), type of discharge, dates of admission and discharge, main diagnosis, comorbid diagnosis, length of stay, and global cost. All the hospitalizations were grouped by age into three categories: 65–74 years old, 75–84 years old and older people (85 years old and above). We excluded readmissions, defined as admissions during the first 30 days after discharge in the same hospital.

## Statistical methods

Quantitative data are described with averages and standard deviations or medians with IQRs (interquartile ranges), and categorical data are expressed with frequencies (absolutes and percentages). Crude rates were defined as hospitalizations per 1,000 inhabitants aged 65 or older, according to population data on each July 1st, provided by the National Statistics Institute of Spain. These data were also used to obtain age-sex standardization rates of both global and specific UTI conditions, which were calculated using the 2015 Spanish population.

To identify possible varying trends over time, we performed a Joinpoint regression [21] using age-adjusted UTI rates, with the 2015 population as the reference population. This widely used approach fits a linear model when Joinpoints (i.e., change points) are not known a priori and are to be estimated from the data. Two measures are provided: an estimation of the annual percent change (APC) in each linear segment and the average annual percent change (AAPC), computed as a weighted average of the APCs from the model. In case no Joinpoints were obtained, the AAPC exactly reflected the APC.

Statistical analysis was performed using Stata 14 and Joinpoint Regression Program, version 4.5.0.1 (National Cancer Institute).

## Ethics statement

The study was approved by the Hospital Fundación Alcorcon Ethics Research Committee (20/125) favorably evaluated by the Central Research Commission of the Primary Healthcare Management of Madrid (57/20). The need for consent was waived by the Ethics Committee (20/125). The CMBD data were fully anonymized by the Ministry of Health before investigators accessed to them.

## Results

During the 2000–2015 period, there were 387,010 hospitalizations among patients over 65 years; 81.15% were diagnosed with non-specified UTI, 0.68% with cystitis, 14.06% with pyelonephritis and 4.10% with prostatitis. The results were adjusted by reviewing non-specified UTI hospitalizations and considering secondary diagnosis. "Fig 1" shows the flow diagram for the hospitalizations.

### Socio-demographic characteristics and clinical features of urinary tract infection-related hospitalizations

Out of 387,010 registered hospitalizations during the study period, 42.41% of them corresponded to the 75–84 years old age group. Relative to the sex category, 56.54% of the total were female. The most frequent comorbidity was diabetes (32.01%), with dementia as the second most common comorbidity (15.50%). The median length of stay was 6 days (IQR: 3–10). The median cost was 3,061 euros (IQR: 2,253–4,079), with non-specified UTIs and cystitis having the highest cost. The overall lethality rate was 5.39%. Conditions with the highest lethality rate were non-specified UTIs (6.19%), followed by pyelonephritis (2.36%) and cystitis (2.31%). Upon discharge from the hospital, 89% of patients returned home and 4% were transferred to another hospital or social health center, where the proportion for patients with non-specified UTIs was higher than for other UTIs (2.05% to 3.3%). All pathologies except for COPD were more prevalent in patients with non-specified UTIs. "Table 1" presents patients' characteristics globally and by different conditions.

### Hospitalization rates by sex and age category and UTI type

The crude rates of UTI hospitalizations during the 2000–2015 period varied from 2.04 in 2000 to 4.33 in 2015. The rate of non-specified UTIs doubled from 1.56 in 2000 to 3.63 in 2015 and the rate of prostatitis, from 0.20 in 2000 to 0.48 in 2015, while the rates of cystitis and pyelonephritis remained more stable. See Tables 2 and 3.

When analyzing hospitalizations by age group, higher rates were observed in patients aged 85 years or older, with a five-fold increase between 2000 and 2015 (Table 2). This trend was seen in all UTI types (Figs 3–5) except for prostatitis-related hospitalizations, in which the hospitalization rate was higher among the 65–74 years old age group; see S2 File (and Fig 6).

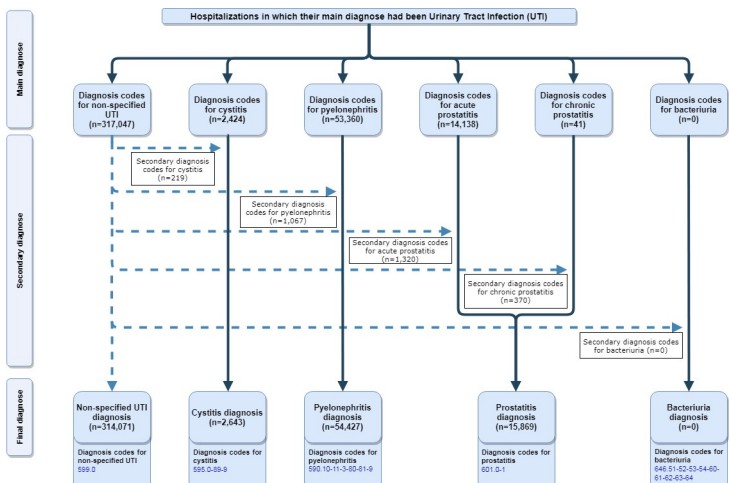

**Fig 1. Flow chart.** Flow diagram for hospitalizations.

**Table 1. Characteristics of urinary tract infection (globally and by different pathologies).**

| | Total | Non-specified UTI | Cystitis | Pyelonephritis | Prostatitis |
|---|---|---|---|---|---|
| | n = 387 010 | n = 314 071 | n = 2 643 | n = 54 427 | n = 15 869 |
| **Age groups*** | | | | | |
| 65–74 years | 97 726 (25.25) | 64 088 (20.41) | 815 (30.84) | 23 709 (43.56) | 9114 (57.43) |
| 75–84 years | 163 376 (42.41) | 134 333 (42.77) | 1 155 (43.70) | 22 452 (41.25) | 5436 (34.26) |
| ≥85 years | 125 908 (32.53) | 115 650 (36.82) | 673 (25.46) | 8 266 (15.19) | 1319 (8.31) |
| **Sex*** | | | | | |
| Male | 168 193 (43.46) | 133 819 (42.61) | 1 283 (48.54) | 17 222 (31.64) | 15 869 (100) |
| Female | 218 804 (56.54) | 180 242 (57.39) | 1 360 (51.46) | 37 202 (68.35) | - |
| **Type of admission*** | | | | | |
| Urgent | 371 536 (96.00) | 301 564 (96.02) | 1 935 (73.21) | 52 657 (96.75) | 15 380 (96.92) |
| Programmed | 15 388 (3.89) | 12 432 (3.96) | 707 (26.75) | 1 761 (3.24) | 488 (3.08) |
| **Comorbidities*** | | | | | |
| Diabetes mellitus | 123 901 (32.01) | 102 949 (32.78) | 785 (29.70) | 16 134 (29.64) | 4 033 (25.41) |
| Dementia | 59 990 (15.50) | 57 047 (18.16) | 251 (9.52) | 2 329 (4.28) | 363 (2.29) |
| Malignant neoplasia | 44 027 (11.38) | 37 781 (12.03) | 283 (10.71) | 4 818 (8.85) | 1 145 (7.22) |
| COPD | 33 568 (8.67) | 27 481 (8.75) | 245 (9.27) | 3 554 (6.53) | 2 288 (14.42) |
| Renal failure | 29 844 (7.71) | 26 140 (8.32) | 187 (7.08) | 2 810 (5.16) | 707 (4.46) |
| Liver disease | 5 674 (1.47) | 4 564 (1.45) | 24 (0.91) | 915 (1.68) | 171 (1.08) |
| **Length of stay \*\*** | 6 (3–10) | 6 (4–10) | 4 (2–8) | 5 (3–9) | 4 (3–6) |
| **Length of stay*** | | | | | |
| 0–3 days | 98 112 (25.35) | 76 481 (24.35) | 1 092 (41.32) | 14 265 (26.21) | 6 274 (39.54) |
| 4–7 days | 147 430 (38.09) | 117 104 (37.29) | 816 (30.87) | 22 572 (41.47) | 6 938 (43.72) |
| 8–11 days | 72 251 (18.67) | 61 024 (19.43) | 336 (12.71) | 9 187 (16.68) | 1 704 (10.74) |
| ≥12 days | 69 217 (17.89) | 59 462 (18.93) | 339 (15.10) | 8 403 (15.44) | 953 (6.01) |
| **Cost (€)\*\*** | 3 061 (2 253–4 079) | 3 129 (2 398–4 079) | 2 890 (2 333–3 518) | 2 540 (2 017–3 257) | 2 231 (1 787–2 351) |
| **Type of discharge*** | | | | | |
| Home | 344 884 (89.12) | 276 344 (87.99) | 2 477 (93.72) | 50 783 (93.30) | 15 280 (96.29) |
| Transfer | 17 160 (4.43) | 14 932 (4.75) | 88 (3.33) | 1 815 (3.33) | 325 (2.05) |
| Others/unknown | 4 112 (1.06) | 3 345 (1.07) | 17 (0.64) | 547 (1.01) | 203 (1.28) |
| **Lethality rate*** | 20 854 (5.39) | 19 450 (6.19) | 61 (2.31) | 1 282 (2.36) | 61 (0.38) |

During that period, UTI hospitalization rates were superior among male vs female patients in all age categories but started to overlap in patients aged >85 years from 2013 (Fig 2); see S2 File. This was seen in all UTI types except for pyelonephritis, which was more common in women; see S2 File (and Fig 5).

### Analysis of trends (joinpoint) globally and by UTI type, sex, and age group

**1. Global.** Joinpoint analysis of UTI-related hospitalizations showed a 4.9% (95% CI 3.2; 6.5) annual increasing trend, with a 5.5% (95% CI 4.7; 6.4) annual percentage change (APC) from 2000 to 2010 (see S3 File).

**2. By UTI type.** Changes in trends by type of infection were not observed regarding non-specified UTI "Fig 3"and pyelonephritis-related hospitalizations "Fig 5" during the study period, however, the other two conditions showed changes in trends: one in cystitis-related hospitalization "Fig 4" rates in 2002 and one in prostatitis-related hospitalizations in 2008 "Fig 6". The APCs for cystitis-related hospitalizations were -12.3% (95%CI -36.6; 21.3) from 2000–2002 and 5.1% (95% CI 3.6; 6.6) from 2002–2015. The APCs for prostatitis-related

**Table 2. Hospitalization rates from 2000 to 2015 globally and by age group.**

| Year | Total | Rates by age group | | |
|---|---|---|---|---|
| | | 65–74 years old | 75–84 years old | >85 years old |
| 2000 | 2.04 | 1.25 | 2.67 | 4.99 |
| 2001 | 2.23 | 1.29 | 2.80 | 5.61 |
| 2002 | 2.17 | 1.27 | 2.69 | 5.39 |
| 2003 | 2.43 | 1.33 | 3.00 | 6.35 |
| 2004 | 2.50 | 1.43 | 3.03 | 6.17 |
| 2005 | 2.60 | 1.43 | 3.11 | 6.60 |
| 2006 | 2.87 | 1.48 | 3.45 | 7.39 |
| 2007 | 2.92 | 1.49 | 3.42 | 7.43 |
| 2008 | 3.19 | 1.54 | 3.67 | 8.42 |
| 2009 | 3.36 | 1.58 | 3.81 | 8.87 |
| 2010 | 3.52 | 1.64 | 3.98 | 9.10 |
| 2011 | 3.68 | 1.65 | 4.13 | 9.59 |
| 2012 | 3.73 | 1.67 | 4.18 | 9.51 |
| 2013 | 3.86 | 1.78 | 4.41 | 9.42 |
| 2014 | 4.01 | 1.84 | 4.48 | 10.08 |
| 2015 | 4.33 | 1.91 | 4.81 | 11.12 |

hospitalizations were 4.9% (95% CI 4.0; 5.7) from 2000–2008 and 9.0% (95% CI 8.2; 9.8) from 2008–2015; see S3 File.

During the study period, two types of infection showed changes in trends: prostatitis-related hospitalizations in 2008 "Fig 6" and cystitis-related hospitalization in 2002 "Fig 4"; see S3 File. The APCs for prostatitis-related hospitalizations were 4.9% (95% CI 4.0; 5.7) annual from 2000–2008 and 9.0% (95% CI 8.2; 9.8) annual from 2008–2015. The APCs for cystitis-related hospitalizations were 5.1% (95% CI 3.6; 6.6) annual from 2002–2015. However, changes were not observed regarding pyelonephritis-related hospitalizations "Fig 5" see S3 File.

**Table 3. Hospitalization rates from 2000 to 2015, by UTI category per year.**

| | Non-specified UTI | Cystitis | Pyelonephritis | Prostatitis |
|---|---|---|---|---|
| 2000 | 1.56 | 0.02 | 0.42 | 0.20 |
| 2001 | 1.71 | 0.02 | 0.42 | 0.20 |
| 2002 | 1.64 | 0.01 | 0.43 | 0.20 |
| 2003 | 1.90 | 0.01 | 0.42 | 0.22 |
| 2004 | 1.93 | 0.02 | 0.46 | 0.22 |
| 2005 | 2.03 | 0.02 | 0.45 | 0.24 |
| 2006 | 2.29 | 0.02 | 0.45 | 0.25 |
| 2007 | 2.35 | 0.02 | 0.44 | 0.27 |
| 2008 | 2.64 | 0.02 | 0.42 | 0.27 |
| 2009 | 2.78 | 0.02 | 0.44 | 0.29 |
| 2010 | 2.92 | 0.02 | 0.44 | 0.32 |
| 2011 | 3.06 | 0.03 | 0.45 | 0.35 |
| 2012 | 3.10 | 0.03 | 0.43 | 0.38 |
| 2013 | 3.21 | 0.02 | 0.45 | 0.41 |
| 2014 | 3.33 | 0.03 | 0.46 | 0.44 |
| 2015 | 3.63 | 0.03 | 0.47 | 0.48 |

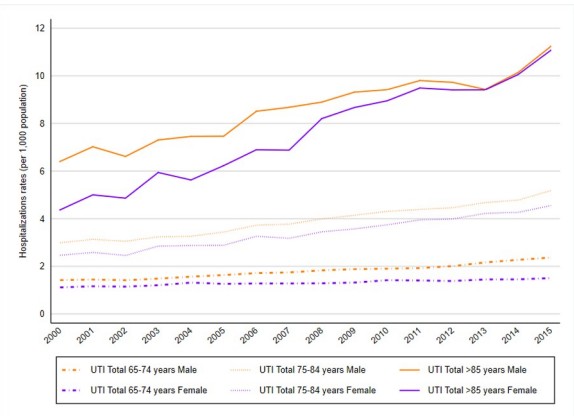

**Fig 2. UTIs total.** UTIs hospitalization rates by sex and group of age (2000–2015).

**3. Temporal analysis by sex.** In the Joinpoint analysis by age groups of UTI hospitalization rates by sex, a statistically significant increasing trend was observed in both sex categories and was superior in the case of female patients vs male patients regarding UTI hospitalization. The AAPC between 2000–2015 in men was 3.8 (95% CI 2.4;5.2) annual, and in women was 4.5 (95% CI 4.1; 4.9) annual "Fig 2"; see S3 File.

The AAPCs from 2000–2015 were 3.9 (95% CI 2.4; 5.5) annual in men and 5.4 (95% CI 4.5; 6.2) annual in women in Non-Specified UTI "Fig 3"; see S3 File.

Regarding cystitis-related hospitalizations, the increasing trend was only statistically significant in male patients, with an AAPC of 3.9 (95% CI 1.7; 6.0) annually (Fig 4); see S3 File.

Concerning pyelonephritis-related hospitalizations from 2000–2015, a decreasing trend annual was observed among male patients, AAPCs from 2000–2015–0.6 (95% CI -1.1; 0.0) while an annual 0.6% (95% CI 0.2;1.0) increasing trend was observed among female patients. There were no changes in trend in either sex during this period (Fig 5); see S3 File.

There was a statistically significant increase, with an AAPC of 6.8 (95% CI 6.2; 7.3) annually, in the incidence rate of prostatitis from 2000–2015 (Fig 6), where the increase was higher after 2008 (AAPC 4.9 [95% CI 4.0; 5.7] versus AAPC 9.0 [95% CI 8.2; 9.8]).

**4. Temporal analysis by age category.** During the study period, the 85 years old and older age group had the highest increasing trend annual relative to UTI hospitalizations APPC

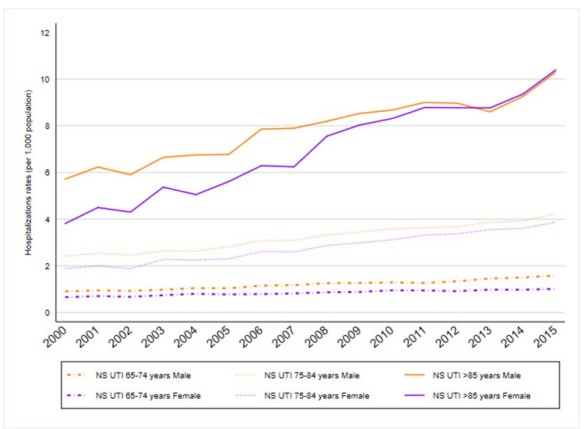

**Fig 3. Non-specified UTIs.** Non-specified UTIs hospitalization rates by sex and group of age (2000–2015).

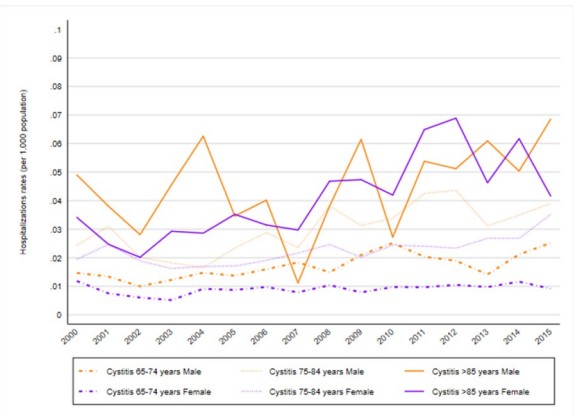

**Fig 4. Cystitis.** Cystitis hospitalization rates by sex and group of age(2000–2015).

5.4% (95% CI 3.5; 7.4), while the 65–74 years old and 75–84 years old age groups had 2.79 (95% CI 2.6; 3.0) and 4.1 (95% CI 3.8; 4.3) increases, respectively; see S4 File.

Non-specified UTI-related hospitalizations trends annual by age category were AAPC 3.4 (95% CI 1.4; 5.0), 65–74 years old, AAPC 4.5 (95% CI 4.2; 4.9) 75–84 years old; and AAPC 5.8 (95% CI 3.6; 7.9) >85 years old age analyses; see S4 File.

Cystitis-related hospitalization trends annual by age, the AAPC from 2000–2015 were 3.2 (95% CI 1.4; 5.0) 65–74 years old, 3.9 (95% CI 2.1; 5.6) 75–84 years old and 4.8 (95% CI 2.3; 7.4) in >85 years old age analyses. There were no changes in the trend; see S4 File.

Trends annual by age category regarding pyelonephritis-related hospitalizations only were statistically significant increase in 75–84 years old age AAPC 0.6 (95% CI 0.2; 0.9).

The annual prostatitis-related hospitalization trends during the study period were less pronounced in the 65–74 years old age group, with an AAPC of 5.5 (95% CI 3.8; 7.2) versus an AAPC of 7.7 (95% CI 6.8; 8.7) in the 75–84 years old age group and an AAPC of 7.2 (95% CI 3.0; 11.6) in the >85 years old age group. The changes in trend were more pronounced in the last few years, from 2003–2004, with an AAPC of 7 (95% CI 6.1; 7.8) annually in the 65–74 years age group and an AAPC of 11.5 (95% CI 9.2; 13.8) annually in the >85 years age group; see S4 File.

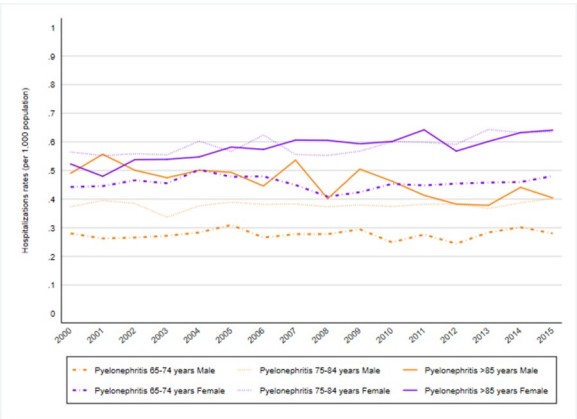

**Fig 5. Pyelonephritis.** Pyelonephritis hospitalization rates by sex and group of age (2000–2015).

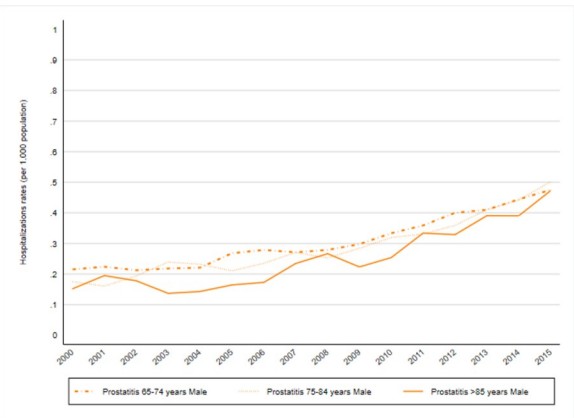

**Fig 6. Prostatitis.** Prostatitis hospitalization rates by group of age (2000–2015).

## Discussion

Our study has analyzed the changes in hospital admissions due to UTIs in Spain over 16 years (2000–2016), confirming the trends in incidence and related factors.

The rates of hospitalization due to UTIs increased with age, particularly in those over 85 years old, and were higher in men than in women for all age groups, tending to equalize in those over 85 years old. Our study included a higher percentage of women, as in other studies [6, 10, 11, 13, 22], but after standardizing by population and sex, the hospitalization rate was higher in men.

More than 80% of the UTI admissions in our study were for non-specified UTIs, with higher rates in men in all age groups throughout the entire study period. Over the period analyzed, hospitalization rates for non-specified UTIs doubled, increasing in all age groups and particularly in those over 85 years old and in both sexes, with a greater annual rate increase in women.

These non-specified UTIs required longer hospital stays, with a higher associated cost and greater lethality. A larger proportion of cases was referred to other hospitals and/or social health centers than for other UTIs. The data showing worse prognosis can be explained by the fact that they are associated with older people with more comorbidities (diabetes, dementia, kidney or liver disease, neoplasia) than other UTIs. One exception is COPD, which was more common among patients admitted with prostatitis, justified by this pathology being associated with male sex.

The higher percentage of non-specified UTIs can be justified by the difficulty of reaching a specific diagnosis in older patients. This group of patients has more non-specific symptoms and a high prevalence of asymptomatic bacteriuria in both patients with community-acquired disease and, in particular, institutionalized patients, patients with urinary catheters, and patients with cognitive deficits, all of which make it more difficult to reach a correct diagnosis [23, 24].

In our study, 14% of admissions were for pyelonephritis. Unlike non-specified UTIs, pyelonephritis was more common in women and was homogeneously distributed over the study period. Hospitalization rates were highest in women aged over 75 years and men aged over 85 years. The associated mortality was 2.3%. In other studies, age was the best predictor of mortality for cases of pyelonephritis that required hospital admission, with mortality rates of 0.7% in women and 1.6% in men [25]. Other factors that affect mortality due to pyelonephritis are immune suppression and the onset of septic shock [26].

In our study, 4% of all UTI admissions were for prostatitis (9.4% of infections in men). Unlike non-specified UTIs, the rate was lower in those aged over 85 years. The rates doubled over the study period, increasing in those over 75 years old. A greater increase (11.4%) occurred from 2004 in those over 85 years old. The global burden of this disease remains unknown. Acute prostatitis affects approximately 1% of males throughout their life, with hospital care sometimes required to prevent adverse outcomes due to sepsis [27]. Age-incidence is bimodal, with a peak at approximately 20–40 years and a second peak after 60 years [28, 29]. In a study of patients treated for prostatitis in hospital emergency departments, admissions were associated with age over 75 years (like in our study's results), a history of urinary tract interventions, neoplasia, prescription of antibiotics in the last three months and the development of bacterial resistance [27].

One possible explanation for the increase in hospitalizations due to UTIs is the increasing antibiotic resistance of community-acquired infections, which reduces the efficacy of current options for outpatient treatment [5, 6]. In this regard, fluoroquinolones are an important factor due to their high resistance rate in all age groups over 50 years old [9, 12]. Overtreatment and incorrect use of antibiotics at the community [30] and hospital level [31] and deficiencies in the social care networks required for community support [32] have also been indicated. One factor that is directly related to the increase in admissions is age [6, 10]. In patients over 85 years old with multiple comorbidities, referral from long-term care facilities [13], delayed administration or delays in starting antibiotic treatment [17, 33] also explain why the course of disease is worse in this age group.

The limitations of the study are related to the limitations of the CMBD. Firstly, this database does not contain any information on the microbiology of the organisms causing UTIs, the associated antibiotic treatment, or the risk factors known to be associated with UTIs. Secondly, it cannot be used to identify subsequent admissions of a specific patient, except those that occur within 30 days of discharge (which were not included in this study).

The results obtained in this study provide an accurate assessment of the changes in the national incidence of hospitalizations due to UTIs over 16 years and allow us to estimate the real burden of UTIs in older patients and their impact on our health system.

Further research is needed on the incidence of UTIs at the community level and on patterns of antibiotic use, which would help to improve the control of resistance and optimize clinical management, establishing clinical pathways for managing this type of infection in older patients. We also need to understand the factors associated with hospitalization in higher-risk patients, such as those with comorbidities or catheters, so we can establish preventive strategies that would reduce admissions and have a favorable impact on patient health and on the health system.

## Supporting information

**S1 File. RECORD checklist (REporting of studies Conducted using Observational Routinely-collected Data).**
(DOCX)

**S2 File. Incidence rates by year, condition, age category, and sex.**
(DOCX)

**S3 File. Trend analysis (Joinpoint) by condition adjusted by age category.**
(DOCX)

**S4 File. Trend analysis (Joinpoint) by condition adjusted by sex category.**
(DOCX)

## Acknowledgments

To our colleagues from the Research Unit for their support: Marcial Caboblanco-Muñoz, Juan Carlos Gil-Moreno, Jaime Barrio-Cortés.

To Dr. María del Canto de Hoyos for her comprehensive revision of the final draft.

## Author Contributions

**Conceptualization:** Jesús Redondo-Sánchez, Isabel del Cura-González, Ángel Gil-de Miguel.

**Data curation:** Laura Díez-Izquierdo, Francisco Rodríguez-Cabrera.

**Formal analysis:** Isabel del Cura-González, Laura Díez-Izquierdo, Ricardo Rodríguez-Barrientos, Francisco Rodríguez-Cabrera.

**Funding acquisition:** Jesús Redondo-Sánchez.

**Methodology:** Isabel del Cura-González, Laura Díez-Izquierdo, Ricardo Rodríguez-Barrientos, Francisco Rodríguez-Cabrera, Elena Polentinos-Castro.

**Resources:** Jesús Redondo-Sánchez.

**Supervision:** Jesús Redondo-Sánchez, Isabel del Cura-González.

**Validation:** Jesús Redondo-Sánchez, Isabel del Cura-González, Laura Díez-Izquierdo, Miguel López-Miguel, Lucas Marina-Ono.

**Writing – original draft:** Jesús Redondo-Sánchez, Isabel del Cura-González, Ricardo Rodríguez-Barrientos, Elena Polentinos-Castro.

**Writing – review & editing:** Jesús Redondo-Sánchez, Isabel del Cura-González, Laura Díez-Izquierdo, Ricardo Rodríguez-Barrientos, Francisco Rodríguez-Cabrera, Elena Polentinos-Castro, Miguel López-Miguel, Lucas Marina-Ono, Laura Llamosas-Falcón, Ángel Gil-de Miguel.

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
