## [Decision Letter · Decision Letter 0]

12 Apr 2021

PONE-D-21-01514

TRENDS IN URINARY TRACT INFECTION HOSPITALIZATION IN ELDERLY PATIENTS IN SPAIN FROM 2000-2015

PLOS ONE

Dear Dr. Rodríguez-Barrientos,

Thank you for submitting your manuscript to PLOS ONE. After careful consideration, we feel that it has merit but does not fully meet PLOS ONE’s publication criteria as it currently stands. Therefore, we invite you to submit a revised version of the manuscript that addresses the points raised during the review process.

The manuscript is clearly written and shows interesting results. However, there are a number of issues that require clarification if the manuscript is to be published. Both the introduction and the discussion require some editing. The introduction should present the hypotheses you wanted to verify and the discussion should be expanded to discuss all the reported findings, as pointed out by both Reviewers. There are also minor issues, which are described in detail by the Reviewers.

The manuscript would also benefit from English editing.

We look forward to receiving your revised manuscript.

Kind regards,

Justyna Gołębiewska

Academic Editor

PLOS ONE

Journal Requirements:

Reviewers' comments:

Reviewer's Responses to Questions

**Comments to the Author**

1. Is the manuscript technically sound, and do the data support the conclusions?

Reviewer #1: Yes

Reviewer #2: Partly

2. Has the statistical analysis been performed appropriately and rigorously? 

Reviewer #1: Yes

Reviewer #2: Yes

3. Have the authors made all data underlying the findings in their manuscript fully available?

Reviewer #1: Yes

Reviewer #2: Yes

4. Is the manuscript presented in an intelligible fashion and written in standard English?

Reviewer #1: Yes

Reviewer #2: No

5. Review Comments to the Author

Reviewer #1: This retrospective cohort study analyzes trends in urinary tract infection hospitalization (cystitis, pyelonephritis, prostatitis and non-specified UTI) among patients over 65 years in Spain from 2000-2015. Findings of increasing trends are consistent with previous findings. A very limited discussion of results is provided. Consider providing more context for findings and what can be done from a clinical and/or or public health perspective (infection control / antibiotic stewardship) to reverse these trends.

Abstract:

Results: Please specify that all AAPCs and APCs are per year. For example, an increase in incidence rates of 4.9% per year should be specified.

Conclusion: Consider adding an interpretation to data from a clinical or public health standpoint. Why do you postulate this trend occurred? What do your findings mean for antibiotic stewardship or infection control?

Consider throughout the manuscript changing the use of the word “elderly” which can be offensive to “older adults”.

First paragraph – Inappropriate diagnosis of UTI that is truly asymptomatic bacteriuria (ASB) is also a major problem and older adults. Consider discussing this in your introduction. For example what proportion of UTI hospitalizations are for actually ASB? A lot of older adults with altered mental status/ and a number of other conditions but diagnoses as having a UTI.

Introduction: “Other risk factors associated with community-acquired UTIs that require hospitalization are male sex, presence of a urinary catheter, and a history of urolithiasis and UTIs in previous months[8].” Please provide discussion of why male sex is a risk factor for community-acquired UTIs that require hospitalization. It is well known that UTIs are more common in women than men at all age categories (as per review cited below). Consider also mentioning the incidence of UTI in older adults in males versus females.

“Over 10% of women older than 65 years of age reported having a UTI within the past 12 months [11]. This number increases to almost 30% in women over the age of 85 years [12]. In a large prospective cohort study of post-menopausal women living in the community, the incidence of UTI was 0.07 per person-year and 0.12 per person-year in older women with diabetes [10]. For men aged 65–74 years, the incidence of UTI is estimated to increase to 0.05 per person-year [9]. In both men and women over the age of 85 years, the incidence of UTI increases substantially. A small cohort study in this age group found the incidence of UTI in women to be 0.13 per person-year and 0.08 per person-year in men [13].”

https://www.ncbi.nlm.nih.gov/pmc/articles/PMC3878051/

“Other associated factors are a high prevalence of comorbidities, instrumentation of the urinary tract, polypharmacy and highly resistant pathogens[9-11].” Consider adding that previous antibiotic use, in particular fluroquinolone use, is a risk factor

Introduction: Consider to focusing the introduction more closely on the rationale for this particular study. Paragraphs 2 and 3 could be moved to the discussion.

Introduction: “In addition, Spain has one of the longest life expectancies in the world, with octogenarian patients being the fastest-growing group[22].” Consider removing this sentence, studying UTI hospitalizations in older adults is important globally and not just in Spain.

Introduction: For that purpose, we have analyzed the Spanish Hospitalization Minimum Data Set (CMBD), a database provided by the Ministry of Health, which includes hospitalization data from both public and private hospitals and has proven its usefulness in previous epidemiological studies[23-25].” Consider moving this information on the CMBD to the methods section.

Methods: Participants and data source- What is NHS?

“a total coverage from all kinds of hospitals” What is meant by “all kinds of hospitals”?

Statistical methods: Consider adding that UTIs were analyzed globally and by specific diagnosis.

Please provide a citation for the methods used for Joinpoint methods.

Results: Overall, a lot of subgroup temporal analyses are presented (by 3 different age groups, 2 sex categories, global UTI diagnosis and 4 non-specified UTI, cystitis, pyelonephritis, and prostatitis. This is overwhelming to the readers, especially since most of the findings were generally similar to the global findings. Consider focusing the results text to the global findings and leave the findings by specific diagnosis for the supplement except for significant findings that are different than the global findings, especially since the global findings are largely driven by the (81.15%) non-specified UTI.

First paragraph: “according to the recommendations for presentations made by the RECORD”. Consider removing this – this is part of the methods and has already been cited.

How many of the 387,010 diagnoses were in unique patients?

Table 1: Only 6 comorbidities were assessed – consider including more comorbidities that are prevalent in this age group and/or are risk factors for UTI or poor outcomes in this age group.

Consider changing Table 2 to a Figure – it is very busy as a Table and difficult to visualize findings.

Please specify that all AAPCs and APCs are per year. For example, an increase in incidence rates of 4.9% per year should be specified.

The APCs for cystitis-related hospitalizations were -12.3% (95%CI -36.6; 21.3) from 2000-2002 and 5.1% (95% CI 3.6; 6.6) from 2002-2015. The APCs for prostatitis-related hospitalizations were 4.9% (95% CI 4.0; 5.7) from 2000-2008 and 9.0% (95% CI 8.2; 9.8) from 2008-2015; see S3 File.

Discussion: “Hospitalizations were more frequent for female patients and in the 75-84 years old age group, however, hospitalization rate was higher in the >85 years old. By sex, the hospitalization rates were higher in men than in women in all age groups except ≥ 85 years.” Consider mentioning overall findings by sex and then by age; combining sex and age groups makes findings hard to follow for readers.

Consider commenting on reasons why rates were higher in males than females? Especially since previous work has demonstrated UTI rates are higher in older females than males.

Very limited discussion of results by condition (two sentences provided for each pyelonephritis and prostatitis results). Consider providing more discussion on findings by condition.

Consider commenting on reasons for the increasing trends observed and what can be done from a clinical and/or or public health perspective (infection control / antibiotic stewardship) to reverse these trends.

Consider commenting further on limitations. For example the data cannot distinguish between true UTI vs ASB and a limited number of clinical characteristics were assessed (for example no information on previous or current antibiotic use, no microbiology data to identify causative organisms, no clinical data).

Reviewer #2: Dear authors, i think this is an interesting paper but it could deserve some english improvements and some clinical clarifications before publication.

INTRODUCTION:

This section is too long, you presented too many data, distracting the reader from the essential points of their research.

Do you have any hypotheses ?

" For that purpose, we have analyzed the Spanish Hospitalization Minimum Data Set (CMBD), a database provided by the

4 Ministry of Health, which includes hospitalization data from both public and private hospitals and has proven its usefulness in previous epidemiological studies [23-25]"

-> I think it is not useful to add this sentence. I think you will present the same concept in the section "methods".

METHODS

"We excluded re-admissions, defined as admissions during the first 30 days after discharge in the same hospital". Then i understand correctly, you did not exclude 're-admissions' in others hospitals. Can you comment this choice ?

RESULTS

It could be interesting to give more informations about others comorbidities (e.g. ischemic heart diseases, HTA) and about the place of living (nursing home, home, others) in the table 1.

You have to clarify the difference between hospitalization "programmed" and "urgent"

-> Although not mandatory it could be interesting to clarify if, speaking about "programmed hospitalizations", they were programmed to treat UTI or UTI was an incident diagnostic during hospitalization due to other reasons (e.g. hip prothesis).

DISCUSSION

In clinical practice is not "easy" to classify urinary infection sin "cystitis", "pyelonephritis" and "prostatitis". Moreover UTI are often a "diagnostic of exclusion" when the clinician is not able to find the source of infection. I think you have to clarify that you have done an "epidemiological exercise" to avoid creating further confusion (in a field that is already very confusing).

In discussion you said: "Other factors that could potentially be related to these observations are higher treatment failure rates linked to antimicrobial resistance[6][11], late onset of treatment in these patients[7] and procedence from long-term care facilities, as well as higher presence of multiple comorbidities among patients in this age group [8]".

-> These sentences are not supported by the data you presented.

The discussion is "poor". I think you can extend it for example by discussing about the economic impact of UTI, and the potential strategies for prevention of UTI.

A final sentence, including the clinical significance of your findings is missing.

OTHER

Avoid the use of the word "elderly"

CONCLUSION

In conclusion, in my opinion the work, although not fully original, it seems to be of some interest and could further expand the knowledge of these infections in the elderly patients, but it has to be revised. Provided that these points are expanded, I would suggest it for publication in the Journal.

6. PLOS authors have the option to publish the peer review history of their article (what does this mean?). If published, this will include your full peer review and any attached files.

Reviewer #1: No

Reviewer #2: **Yes: **Davide Angioni

---

## [Author Response · Author response to Decision Letter 0]

15 Jul 2021

Dear Editor:

Thank you for the thorough review of our manuscript. All of the comments have been addressed and the manuscript is much improved. Please see our responses to the reviewer’s comments below.

Editor comments:

Thank you for your comment. We have revised our submission according to the PLOS ONE style template.

Thank you for your comment. We have corrected the information on ‘Funding Information’ and ‘Financial Disclosure’ sections to match. The project was funded by the Instituto de Salud Carlos III, PI19/01700 (Spanish Ministry of Science and Innovation). In addition, the principal investigator JRS received support to increase his research activities and to publish this manuscript from the 2020 funding program of the Fundación de Investigación e Innovación Biomédica en Atención Primaria (FIIBAP), Community of Madrid.

3. We note that you have indicated that data from this study are available upon request. PLOS only allows data to be available upon request if there are legal or ethical restrictions on sharing data publicly. In your revised cover letter, please address the following prompts:

The CMBD data belong to the Ministry of Health and are partially accessible to the public. For data series on hospitalization, like the data presented in this article, specific selections of anonymized microdata from CMBD records can be requested from the ministry.

The application form is at https://www.mscbs.gob.es/estadEstudios/estadisticas/estadisticas/estMinisterio/SolicitudCMBD.htm

https://www.mscbs.gob.es/estadEstudios/estadisticas/estadisticas/estMinisterio/SolicitudCMBD.htm

The patient information was anonymized and de-identified prior to the analysis. Ethics Committee (Comité de Ética del Hospital Fundación Alcorcón) ruled that no formal ethics approval was required in this particular case.

b) if there are no restrictions, please upload the minimal anonymized data set necessary to replicate your study findings as either Supporting Information files or to a stable, public repository and provide us with the relevant URLs, DOIs, or accession numbers. Please see http://www.bmj.com/content/340/bmj.c181.long for guidelines on how to de-identify and prepare clinical data for publication. For a list of acceptable repositories, please see

See the previous response.

The organization that manages research within Primary Health Care Management is the Fundación de Investigación e Innovación Biomédica en Atención Primaria. All the investigators are affiliated with this foundation. If you wish to confirm this, you can contact fiibap@salud.madrid.org or check the website.

Reviewer comments:

Reviewer #1:

This retrospective cohort study analyzes trends in urinary tract infection hospitalization (cystitis, pyelonephritis, prostatitis and non-specified UTI) among patients over 65 years in Spain from 2000-2015. Findings of increasing trends are consistent with previous findings. A very limited discussion of results is provided. Consider providing more context for findings and what can be done from a clinical and/or public health perspective (infection control / antibiotic stewardship) to reverse these trends.

Thank you very much for your comment. We are grateful for the opportunity to add this information to the discussion section, including the reviewer’s suggestions. They have been added to the discussion section on page 9, paragraphs 10, page 10, paragraphs 4 and 6, and page 11, paragraphs 1

Regarding the suggestion “Consider providing more context for findings”:

The higher percentage of non-specified UTIs can be justified by the difficulty of reaching a specific diagnosis in older patients. This group of patients has more non-specific symptoms and a high prevalence of asymptomatic bacteriuria in both patients with community-acquired disease and, in particular, institutionalized patients, patients with urinary catheters, and patients with cognitive deficits, all of which make it more difficult to reach a correct diagnosis (Pescatore et al. 2019) (Rowe and Juthani-Mehta 2013).

One possible explanation for the increase in hospitalizations due to UTIs is the increasing antibiotic resistance of community-acquired infections, which reduces the efficacy of current options for outpatient treatment (Simmering et al. 2017) (Zilberberg and Shorr 2013). In this regard, fluoroquinolones are an important factor due to their high resistance rate in all age groups over 50 years old (Goldstein et al. 2019) (Ahmed et al. 2019). Overtreatment and incorrect use of antibiotics at the community (Bruxvoort et al. 2020) and hospital level (Palacios-Ceña et al. 2017) and deficiencies in the social care networks required for community support (Rosello et al. 2018) have also been indicated. One factor that is directly related to the increase in admissions is age (Simmering et al. 2017) (Caljouw et al. 2011). In patients over 85 years old with multiple comorbidities, referral from long-term care facilities (Medina-Polo et al. 2015), delayed administration or delays in starting antibiotic treatment (Gharbi et al. 2019) (Álvarez Artero et al. 2019) also explain why the course of disease is worse in this age group.

Regarding the suggestion “Consider providing what can be done from a clinical and/or public health perspective (infection control / antibiotic stewardship) to reverse these trends.”

The results obtained in this study provide an accurate assessment of the changes in the national incidence of hospitalizations due to UTIs over 16 years and allow us to estimate the real burden of UTIs in older patients and their impact on our health system

We also need to understand the factors associated with hospitalization in higher-risk patients, such as those with comorbidities or catheters, so we can establish preventive strategies that would reduce admissions and have a favorable impact on patient health and on the health system.

Abstract:

Results: Please specify that all AAPCs and APCs are per year. For example, an increase in incidence rates of 4.9% per year should be specified.

The references to the joinpoint results have been amended in the abstract and throughout the document to emphasize that these are annual trends. The text in the abstract (page 2) reads as follows:

Joinpoint analyses showed an average annual percentage increase (AAPC) in incidence rates of 4.9% (95% CI 3.2;6.5) in UTI hospitalizations, 4.6% (95% CI 4.2;5.1) in non-specified UTI and 6.8% (95% CI 6.2; 7.3) annual in prostatitis.

Conclusion: Consider adding an interpretation to data from a clinical or public health standpoint. Why do you postulate this trend occurred? What do your findings mean for antibiotic stewardship or infection control?

Thank you for your comment. We have incorporated these aspects into the abstract’s conclusion and the discussion section. 

Consider throughout the manuscript changing the use of the word “elderly” which can be offensive to “older adults”.

Thank you very much. We have replaced “elderly” with “older adults” in the text and title

First paragraph – Inappropriate diagnosis of UTI that is truly asymptomatic bacteriuria (ASB) is also a major problem and older adults. Consider discussing this in your introduction. For example what proportion of UTI hospitalizations are for actually ASB? A lot of older adults with altered mental status/ and a number of other conditions but diagnoses as having a UTI.

We have included these issues in the discussion. The text in the discussion section (manuscript page 9, paragraph 10) reads as follows: 

The higher percentage of non-specified UTIs can be justified by the difficulty of reaching a specific diagnosis in older patients. This group of patients has more non-specific symptoms and a high prevalence of asymptomatic bacteriuria in both patients with community-acquired disease and, in particular, institutionalized patients, patients with urinary catheters, and patients with cognitive deficits, all of which make it more difficult to reach a correct diagnosis (Pescatore et al. 2019) (Rowe and Juthani-Mehta 2013).

Introduction: “Other risk factors associated with community-acquired UTIs that require hospitalization are male sex, presence of a urinary catheter, and a history of urolithiasis and UTIs in previous months [8].” Please provide discussion of why male sex is a risk factor for community-acquired UTIs that require hospitalization. It is well known that UTIs are more common in women than men at all age categories (as per review cited below). Consider also mentioning the incidence of UTI in older adults in males versus females.

“Over 10% of women older than 65 years of age reported having a UTI within the past 12 months [11]. This number increases to almost 30% in women over the age of 85 years [12]. In a large prospective cohort study of post-menopausal women living in the community, the incidence of UTI was 0.07 per person-year and 0.12 per person-year in older women with diabetes [10]. For men aged 65–74 years, the incidence of UTI is estimated to increase to 0.05 per person-year [9]. In both men and women over the age of 85 years, the incidence of UTI increases substantially. A small cohort study in this age group found the incidence of UTI in women to be 0.13 per person-year and 0.08 per person-year in men [13].”

https://www.ncbi.nlm.nih.gov/pmc/articles/PMC3878051/

Thank you for your suggestions. We have included these issues in the introduction. We have not added the causes because we do not know them. They could be due to the catheters, but this should be confirmed by additional research studies. The text in the introduction section (manuscript page 3, paragraph 3) reads as follows:

Previous community-level population studies have revealed an increasing trend of UTIs with age (Ahmed et al. 2018), reporting a higher risk of bacteremia and death by all causes in male patients and patients above 85 years, especially in depressed areas (Gharbi et al. 2019).

Other associated factors are a high prevalence of comorbidities, instrumentation of the urinary tract, polypharmacy and highly resistant pathogens [9-11].” Consider adding that previous antibiotic use, in particular fluroquinolone use, is a risk factor

These issues, including the subject of antibiotic and fluoroquinolone use, have been added to the introduction (page 3, paragraph 2) and the discussion (page 10, paragraph 4). The text reads as follows:

In the introduction:

In recent years, hospital admissions due to UTIs have increased by up to 50% (Zilberberg and Shorr 2013) (Simmering et al. 2017) (Blunt I s. f.). The risk factors associated with community-acquired UTIs that require hospitalization include male sex (Ahmed et al. 2019b), high prevalence of comorbidities such as dementia, stroke or urinary incontinence (Caljouw et al. 2011), instrumentation of the urinary tract, polypharmacy and highly resistant pathogens associated with treatment failure (Eliakim-Raz et al. 2018) (Goldstein et al. 2019).

In the discussion:

One possible explanation for the increase in hospitalizations due to UTIs is the increasing antibiotic resistance of community-acquired infections, which reduces the efficacy of current options for outpatient treatment (Simmering et al. 2017) (Zilberberg and Shorr 2013). In this regard, fluoroquinolones are an important factor due to their high resistance rate in all age groups over 50 years old (Goldstein et al. 2019) (Ahmed et al. 2019). Overtreatment and incorrect use of antibiotics at the community (Bruxvoort et al. 2020) and hospital level (Palacios-Ceña et al. 2017)

Introduction: Consider to focusing the introduction more closely on the rationale for this particular study. Paragraphs 2 and 3 could be moved to the discussion.

Thank you. We have moved the previous paragraphs 2 and 3, on pyelonephritis and prostatitis, to the discussion as you suggest. The introduction now focuses on UTIs in general and on hospitalizations due to UTIs, without including information on categories or subgroups of infection.

Introduction: “In addition, Spain has one of the longest life expectancies in the world, with octogenarian patients being the fastest-growing group [22].” Consider removing this sentence, studying UTI hospitalizations in older adults is important globally and not just in Spain.

Thank you for your suggestion. We have removed this sentence.

Introduction: For that purpose, we have analyzed the Spanish Hospitalization Minimum Data Set (CMBD), a database provided by the Ministry of Health, which includes hospitalization data from both public and private hospitals and has proven its usefulness in previous epidemiological studies [23-25].” Consider moving this information on the CMBD to the methods section.

We have made the changes suggested by the reviewer (paragraph, methods section), adding, when discussing the CMBD database, its definition of hospitalization and readmission criteria.

The text in the methods section (page 3, paragraph 7) reads as follows:

The CMBD is a registry for all hospitals in Spain that contains data for up to 97.7% of discharges from public hospitals during all periods, and from 2005, the CMBD had gradual coverage from private hospitals, reaching a total coverage from all kinds of hospitals of 92% in 2014. The CMBD defines hospitalization as admission to the emergency department for over 24 hours and/or scheduled admission. Admissions that occur within 30 days of discharge are considered to be readmissions. The CMBD has proven its usefulness in previous epidemiological studies [19] [20].

Methods: Participants and data source- What is NHS?

The definition has been added to the Participants and data source section: Spanish National Health Service (SNHS).

Additional information on the Spanish National Health Service is available at: http://www.msps.es/organizacion/sns/librosSNS.htm

“a total coverage from all kinds of hospitals” What is meant by “all kinds of hospitals”? The CMBD is a registry for all hospitals in Spain that contains data for up to 97.7% of discharges from public hospitals during all periods, and from 2005, the CMBD had gradual coverage from private hospitals, reaching a total coverage from all kinds of hospitals of 92% in 2014.

The phrase “all kinds of hospitals” has been amended to remove “all kinds”. We were referring to all hospitals in Spain, both public and private. In Spain, hospitals are classified into three levels according to their capacity to treat more complex patients. The CMBD includes all three levels. The text in the methods section (page 3, paragraph 7) reads as follows:

The CMBD is a registry for all hospitals in Spain that contains data for up to 97.7% of discharges from public hospitals during all periods, and from 2005, the CMBD had gradual coverage from private hospitals, reaching a total coverage from all hospitals of 92% in 2014.

Statistical methods: Consider adding that UTIs were analyzed globally and by specific diagnosis.

This has been added as suggested by the reviewer. The text in the statistical analysis section (page 4, paragraph 4) reads as follows:

“These data were also used to obtain age-sex standardization rates of both global and specific UTI conditions, which were calculated using the 2015 Spanish population.”

Please provide a citation for the methods used for Joinpoint methods.

The following reference for the joinpoint method has been added to the methods section:

Kim HJ, Fay MP, Feuer EJ, Midthune DN (2000) Permutation tests for joinpoint regression with applications to cancer rates. Stat Med 19: 335–351.

Results: Overall, a lot of subgroup temporal analyses are presented (by 3 different age groups, 2 sex categories, global UTI diagnosis and 4 non-specified UTI, cystitis, pyelonephritis, and prostatitis. This is overwhelming to the readers, especially since most of the findings were generally similar to the global findings. Consider focusing the results text to the global findings and leave the findings by specific diagnosis for the supplement except for significant findings that are different than the global findings, especially since the global findings are largely driven by the (81.15%) non-specified UTI.

Thank you very much for your suggestion. We have shortened the comments on the results, leaving only the most important comments on each section, emphasizing those that differ from the overall trend (e.g. prostatitis) and removing the other, less interesting analyses, which can be seen in the tables or figures.

Example of a shortened paragraph with some changes in wording:

Regarding the different types of urinary tract infections, hospitalization rates during the studied period increased from a minimum in the year 2000 to a maximum in 2015. Rates per 1,000 inhabitants showed an increase in non-specified UTI hospitalizations from 1.56 in 2000 to 3.63 in 2015, while cystitis-related hospitalizations ranged from 0.02 in 2000 to 0.03 in 2015. Regarding pyelonephritis-related hospitalizations, rates ranged from 0.42 in 2000 to 0.47 in 2015; relative to prostatitis, rates ranged from 0.20 in 2000 to 0.48 in 2015; see S2 File.

This has been changed to:

The rate of non-specified UTIs doubled from 1.56 in 2000 to 3.63 in 2015 and the rate of prostatitis, from 0.20 in 2000 to 0.48 in 2015, while the rates of cystitis and pyelonephritis remained more stable.

Examples of shortened paragraphs:

Regarding cystitis-related hospitalizations, the increasing trend was only statistically significant in male patients, with an AAPC of 3.9 (95% CI 1.7; 6.0) annually (Fig. 4); see S3 File. Non-statistically significant increase AAPC was 2.6 (95% CI -1.5; 6.8) in incidence rate from to cystitis 2000-2015. Non-statistically significant increase AAPCs from 2000-2015 were 3.9 (95% CI 1.7; 6.0) in men and 2.4 (95% CI -1.7; 6.7) in women.

Trends annual by age category regarding pyelonephritis-related hospitalizations only were statistically significant increase in 75-84 years old age AAPC 0.6 (95% CI 0.2; 0.9). two changes in trend were observed in 2010 and 2013 that were not found when analyzing the global trend in hospitalizations;

First paragraph: “according to the recommendations for presentations made by the RECORD”. Consider removing this – this is part of the methods and has already been cited.

Thank you. We have done so. We removed this text as it is already mentioned in the methodology.

How many of the 387,010 diagnoses were in unique patients?

We analyzed UTI episodes, not patients with UTIs. The CMBD database only recognizes admissions of the same patient within 30 days of discharge. We have clarified this point in the methods and discussion sections, as one of the possible limitations of the study.

Methods: page 3, paragraph 7

The CMBD defines hospitalization as admission to the emergency department for over 24 hours and/or scheduled admission. Admissions that occur within 30 days of discharge are considered to be readmissions.

Discussion: page 10, paragraph 5

Secondly, it (CMBD) cannot be used to identify subsequent admissions of a specific patient, except those that occur within 30 days of discharge (which were not included in this study). 

Table 1: Only 6 comorbidities were assessed – consider including more comorbidities that are prevalent in this age group and/or are risk factors for UTI or poor outcomes in this age group.

The CMBD records one primary diagnosis and up to a maximum of 14 secondary diagnoses. These are diagnoses on admission, which are not necessarily the same as the patient’s comorbidities. We have selected the most prevalent diagnoses, which mostly match those previously analyzed in other trend analysis studies.

Consider changing Table 2 to a Figure – it is very busy as a Table and difficult to visualize findings.

Thank you for the suggestion. Figure 2 as submitted previously already shows the distribution of total UTIs by sex in the three age groups, so this information was duplicated in the previous version of table 2. We have simplified the table, leaving only the global data, which are not shown in the figures (the general distribution per year by age group 

Table 2. Hospitalization rates from 2000 to 2015 by age group

 Rates by age group

Year Total 65-74 years old 75-84 years old >85 years old

2000 2.04 1.25 2.67 4.99

2001 2.23 1.29 2.80 5.61

2002 2.17 1.27 2.69 5.39

2003 2.43 1.33 3.00 6.35

2004 2.50 1.43 3.03 6.17

2005 2.60 1.43 3.11 6.60

2006 2.87 1.48 3.45 7.39

2007 2.92 1.49 3.42 7.43

2008 3.19 1.54 3.67 8.42

2009 3.36 1.58 3.81 8.87

2010 3.52 1.64 3.98 9.10

2011 3.68 1.65 4.13 9.59

2012 3.73 1.67 4.18 9.51

2013 3.86 1.78 4.41 9.42

2014 4.01 1.84 4.48 10.08

2015 4.33 1.91 4.81 11.12

The information removed from table 2 (the numerical data on the distribution of total UTIs by sex within each age group) is now included in supplement 3.

Supplement 3 (only the new section on total UTIs is shown here; the rest of the supplement is similar to the previous submission, except for the “totals by age” column for each type of UTI, which is now included in the article as a new table 3).

 Total UTIs

 65-74 75-84 >85 

 M W M W M W

2000 1.42 1.11 2.99 2.46 6.41 4.37

2001 1.44 1.16 3.13 2.58 7.02 5.00

2002 1.42 1.15 3.05 2.45 6.61 4.86

2003 1.48 1.20 3.24 2.85 7.31 5.94

2004 1.56 1.31 3.26 2.87 7.46 5.63

2005 1.63 1.26 3.44 2.88 7.47 6.23

2006 1.71 1.28 3.73 3.26 8.51 6.90

2007 1.74 1.28 3.77 3.18 8.68 6.88

2008 1.83 1.28 3.99 3.45 8.90 8.2

2009 1.88 1.32 4.14 3.57 9.32 8.67

2010 1.90 1.42 4.31 3.74 9.42 8.95

2011 1.93 1.40 4.39 3.95 9.80 9.49

2012 2.01 1.38 4.46 3.99 9.73 9.41

2013 2.16 1.45 4.68 4.23 9.43 9.41

2014 2.27 1.45 4.78 4.27 10.14 10.05

2015 2.37 1.50 5.18 4.55 11.23 11.07

Table 3 Hospitalization rates from 2000 to 2015, by UTI category per year (new)

 Non-specified UTI Cystitis Pyelonephritis Prostatitis

2000 1.56 0.02 0.42 0.20

2001 1.71 0.02 0.42 0.20

2002 1.64 0.01 0.43 0.20

2003 1.90 0.01 0.42 0.22

2004 1.93 0.02 0.46 0.22

2005 2.03 0.02 0.45 0.24

2006 2.29 0.02 0.45 0.25

2007 2.35 0.02 0.44 0.27

2008 2.64 0.02 0.42 0.27

2009 2.78 0.02 0.44 0.29

2010 2.92 0.02 0.44 0.32

2011 3.06 0.03 0.45 0.35

2012 3.10 0.03 0.43 0.38

2013 3.21 0.02 0.45 0.41

2014 3.33 0.03 0.46 0.44

2015 3.63 0.03 0.47 0.48

Please specify that all AAPCs and APCs are per year. For example, an increase in incidence rates of 4.9% per year should be specified. The APCs for cystitis-related hospitalizations were -12.3% (95%CI -36.6; 21.3) from 2000-2002 and 5.1% (95% CI 3.6; 6.6) from 2002-2015. The APCs for prostatitis-related hospitalizations were 4.9% (95% CI 4.0; 5.7) from 2000-2008 and 9.0% (95% CI 8.2; 9.8) from 2008-2015; see S3 File.

The wording has been amended in all parts of the text that discuss this topic (section “Analysis of trends (joinpoint) globally and by UTI type, sex and age group”). Examples: (pp 8-9)

Joinpoint analysis of UTI-related hospitalizations showed a 4.9% (95% CI 3.2; 6.5) annual increasing trend, with a 5.5% (95% CI 4.7; 6.4) annual percentage change (APC) from 2000 to 2010 (see S3 File).

The APCs for prostatitis-related hospitalizations were 4.9% (95% CI 4.0; 5.7) annual from 2000-2008 and 9.0% (95% CI 8.2; 9.8) annual from 2008-2015. The APCs for cystitis-related hospitalizations were 5.1% (95% CI 3.6; 6.6) annual from 2002-2015.

The AAPC between 2000-2015 in men was 3.8 (95% CI 2.4;5.2) annual, and in women was 4.5 (95% CI 4.1; 4.9) annual “Fig 2”; see S3 File.

During the study period, the 85 years old and older age group had the highest increasing trend annual relative to UTI hospitalizations APPC 5.4% (95% CI 3.5; 7.4), while the 65-74 years old and 75-84 years old age groups had 2.79 (95% CI 2.6; 3.0) and 4.1 (95% CI 3.8; 4.3) increases, respectively; see S4 File

Discussion:

“Hospitalizations were more frequent for female patients and in the 75-84 years old age group, however, hospitalization rate was higher in the >85 years old. By sex, the hospitalization rates were higher in men than in women in all age groups except ≥ 85 years.” Consider mentioning overall findings by sex and then by age; combining sex and age groups makes findings hard to follow for readers.

Thank you for your comment. The combined commentary on age and sex has been removed from the text for ease of reading. It reads as follows: page 9, paragraph 7

The rates of hospitalization due to UTIs increased with age, particularly in those over 85 years old, and were higher in men than in women for all age groups, tending to equalize in those ≥85 years old.

Consider commenting on reasons why rates were higher in males than females? Especially since previous work has demonstrated UTI rates are higher in older females than males.

Thank you for your comment. Although there was a higher percentage of women admitted, after adjusting for the population to calculate the rates, these rates were higher in men, except for pyelonephritis, which had higher rates in women, as can be seen in the figures and in supplement 3. This is mainly due to the group of non-specified UTIs and may be related to infections associated with catheters or other instruments more frequently used in men. The text now reads as follows: page 9, paragraph 7

Our study included a higher percentage of women, as in other studies (Simmering et al. 2017) (Eliakim-Raz et al. 2018) (Spoorenberg et al. 2014) (Medina-Polo et al. 2015) (Caljouw et al. 2011), but after standardizing by population and sex, the hospitalization rate was higher in men.

Very limited discussion of results by condition (two sentences provided for each pyelonephritis and prostatitis results). Consider providing more discussion on findings by condition.

Thank you for your comment. Following the recommendations, we have expanded the discussion of results by condition, moving some text from the introduction to the discussion and amending the wording. The text in the discussion section (page 9, paragraphs 8 ,9 and 10 and page 10, paragraphs 2-3) reads as follows:

More than 80% of the UTI admissions in our study were for non-specified UTIs, with higher rates in men in all age groups throughout the entire study period. Over the period analyzed, hospitalization rates for non-specified UTIs doubled, increasing in all age groups and particularly in those ≥85 years old and in both sexes, with a greater annual rate increase in women.

These non-specified UTIs required longer hospital stays, with a higher associated cost and greater lethality. A larger proportion of cases was referred to other hospitals and/or social health centers than for other UTIs. The data showing worse prognosis can be explained by the fact that they are associated with older people with more comorbidities (diabetes, dementia, kidney or liver disease, neoplasia) than other UTIs. One exception is COPD, which was more common among patients admitted with prostatitis (justified by this pathology being associated with male sex).

The higher percentage of non-specified UTIs can be justified by the difficulty of reaching a specific diagnosis in older patients. This group of patients has more non-specific symptoms and a high prevalence of asymptomatic bacteriuria in both patients with community-acquired disease and, in particular, institutionalized patients, patients with urinary catheters, and patients with cognitive deficits, all of which make it more difficult to reach a correct diagnosis (Pescatore et al. 2019) (Rowe and Juthani-Mehta 2013)

In our study, 14% of admissions were for pyelonephritis. Unlike non-specified UTIs, pyelonephritis was more common in women and was homogeneously distributed over the study period. Hospitalization rates were highest in women aged over 75 years and men aged over 85 years. The associated mortality was 2.3%. In other studies, age was the best predictor of mortality for cases of pyelonephritis that required hospital admission, with mortality rates of 0.7% in women and 1.6% in men (Foxman, Klemstine, and Brown 2003). Other factors that affect mortality due to pyelonephritis are immune suppression and the onset of septic shock (Buonaiuto et al. 2014)

In our study, 4% of all UTI admissions were for prostatitis (9.4% of infections in men). Unlike non-specified UTIs, the rate was lower in those aged over 85 years. The rates doubled over the study period, increasing in those over 75 years old. A greater increase (11.4%) occurred from 2004 in those over 85 years old. The global burden of this disease remains unknown. Acute prostatitis affects approximately 1% of males throughout their life, with hospital care sometimes required to prevent adverse outcomes due to sepsis (Ferré, Llopis, and Jacob 2016). Age-incidence is bimodal, with a peak at approximately 20-40 years and a second peak after 60 years (Gill and Shoskes 2016) (Brede and Shoskes 2011). In a study of patients treated for prostatitis in hospital emergency departments, admissions were associated with age over 75 years (like in our study’s results), a history of urinary tract interventions, neoplasia, prescription of antibiotics in the last three months and the development of bacterial resistance (Ferré Losa et al. 2019).

Consider commenting on reasons for the increasing trends observed and what can be done from a clinical and/or public health perspective (infection control / antibiotic stewardship) to reverse these trends.

Following the recommendations, we have expanded the discussion with an explanation of the increasing trend of UTIs and our view of the possible implications for clinical and public health. The text in the discussion section (page 10, paragraph 4 and paragraph 6-7) reads as follows:

One possible explanation for the increase in hospitalizations due to UTIs is the increasing antibiotic resistance of community-acquired infections, which reduces the efficacy of current options for outpatient treatment (Simmering et al. 2017) (Zilberberg and Shorr 2013). In this regard, fluoroquinolones are an important factor due to their high resistance rate in all age groups over 50 years old(Goldstein et al. 2019) (Ahmed et al. 2019). Overtreatment and incorrect use of antibiotics at the community (Bruxvoort et al. 2020) and hospital level (Palacios-Ceña et al. 2017) and deficiencies in the social care networks required for community support (Rosello et al. 2018) have also been indicated. One factor that is directly related to the increase in admissions is age (Simmering et al. 2017) (Caljouw et al. 2011). In patients over 85 years old with multiple comorbidities, referral from long-term care facilities (Medina-Polo et al. 2015), delayed administration or delays in starting antibiotic treatment (Gharbi et al. 2019) (Álvarez Artero et al. 2019) also explain why the course of disease is worse in this age group.

The results obtained in this study provide an accurate assessment of the changes in the national incidence of hospitalizations due to UTIs over 16 years and allow us to estimate the real burden of UTIs in older patients and their impact on our health system.

Further research is needed on the incidence of UTIs at the community level and on patterns of antibiotic use, which would help to improve the control of resistance and optimize clinical management, establishing clinical pathways for managing this type of infection in older patients. We also need to understand the factors associated with hospitalization in higher-risk patients, such as those with comorbidities or catheters, so we can establish preventive strategies that would reduce admissions and have a favorable impact on patient health and on the health system.

Consider commenting further on limitations. For example the data cannot distinguish between true UTI vs ASB and a limited number of clinical characteristics were assessed (for example no information on previous or current antibiotic use, no microbiology data to identify causative organisms, no clinical data).

Thank you for the suggestion. These limitations are related to the use of the database and its limitations. We have added this to the discussion on limitations. The text in the discussion section (page 10, paragraph 5) reads as follows:

The limitations of the study are related to the limitations of the CMBD. Firstly, this database does not contain any information on the microbiology of the organisms causing UTIs, the associated antibiotic treatment, or the risk factors known to be associated with UTIs

Reviewer #2:

Dear authors, I think this is an interesting paper but it could deserve some English improvements and some clinical clarifications before publication.

INTRODUCTION:

This section is too long, you presented too many data, distracting the reader from the essential points of their research.

Thank you for your comment. We have made some changes to highlight the essential points of the research, focusing on total UTIs and hospitalizations and removing the content related to types of UTI. See the introduction section, page 3, paragraphs 1-4.

Do you have any hypotheses?

Our hypothesis was that there was an increase in admissions for total UTIs and for each type of UTI, in both sexes, but that this was more evident in older people due to the aging population???

Thank you. We add it. Considering the aging population, we would expect an increase in total UTIs and in each type of UTI over the study period. 

" For that purpose, we have analyzed the Spanish Hospitalization Minimum Data Set (CMBD), a database provided by the Ministry of Health, which includes hospitalization data from both public and private hospitals and has proven its usefulness in previous epidemiological studies [23-25]"

-> I think it is not useful to add this sentence. I think you will present the same concept in the section "methods".

Thank you. We have removed this paragraph according to your comment.

METHODS

"We excluded readmissions, defined as admissions during the first 30 days after discharge in the same hospital". Then I understand correctly, you did not exclude 'readmissions' in others hospitals. Can you comment this choice?

Thank you for the comment. The text may be confusing. We have corrected it. The CMBD database defines readmission as cases where a patient is admitted again within 30 days. In Spain, this is usually at the same hospital because each patient has a public hospital of reference, but it could be a different hospital, e.g. for people using a mixture of public and private systems. Once 30 days have passed, admissions for UTIs were counted as new admissions, and we cannot exclude the possibility that some were repeat admissions by the same patient, independently of the admitting hospital. Page 3, paragraph 7 reads as follows:

The CMBD defines hospitalization as admission to the emergency department for over 24 hours and/or scheduled admission. Admissions that occur within 30 days of discharge are considered to be readmissions.

RESULTS

It could be interesting to give more information about others comorbidities (e.g. ischemic heart diseases, HTA) and about the place of living (nursing home, home, others) in the table 1.

The CMBD records one primary diagnosis and up to a maximum of 14 secondary diagnoses. These are diagnoses on admission, which are not necessarily the same as the patient’s comorbidities. We have selected the most prevalent diagnoses, which mostly match those previously analyzed in other trend analysis studies. Regarding the place of living, we agree with the reviewer that it would have been very interesting to have data on whether the patient lived in a nursing home or at home, but this information is not provided in the database we were using. 

You have to clarify the difference between hospitalization "programmed" and "urgent"

-> Although not mandatory it could be interesting to clarify if, speaking about "programmed hospitalizations", they were programmed to treat UTI or UTI was an incident diagnostic during hospitalization due to other reasons (e.g. hip prothesis).

Most cases were urgent admissions (96%), meaning that they were admitted via the emergency department for any type of UTI, where this was the primary diagnosis. This is what we wanted to highlight in the article. Scheduled hospitalizations are usually admissions for diagnostic or therapeutic procedures that may or may not be related to UTIs. We cannot exclude the possibility that UTI was the primary diagnosis, but we cannot confirm this using the database, so we have not made any comments in this regard in the results section.

DISCUSSION

In clinical practice is not "easy" to classify urinary infection sin "cystitis", "pyelonephritis" and "prostatitis". Moreover UTI are often a "diagnostic of exclusion" when the clinician is not able to find the source of infection. I think you have to clarify that you have done an "epidemiological exercise" to avoid creating further confusion (in a field that is already very confusing).

We agree, and we are grateful that this comment has given us the opportunity to justify the high frequency of non-specified UTIs more precisely.

We have included the following text in the discussion (page 9, paragraph 10):

The higher percentage of non-specified UTIs can be justified by the difficulty of reaching a specific diagnosis in older patients. This group of patients has more non-specific symptoms and a high prevalence of asymptomatic bacteriuria in both patients with community-acquired disease and, in particular, institutionalized patients, patients with urinary catheters, and patients with cognitive deficits, all of which make it more difficult to reach a correct diagnosis (Pescatore et al. 2019) (Rowe and Juthani-Mehta 2013)

In discussion you said: "Other factors that could potentially be related to these observations are higher treatment failure rates linked to antimicrobial resistance [6][11], late onset of treatment in these patients [7] and procedence from long-term care facilities, as well as higher presence of multiple comorbidities among patients in this age group [8]".

-> These sentences are not supported by the data you presented.

Thank you for the suggestion. We have changed the wording of this sentence because it was interpreted as a result and we intended it as an explanation. The text in the discussion section (page 10, paragraph 4) reads as follows:

One possible explanation for the increase in hospitalizations due to UTIs is the increasing antibiotic resistance of community-acquired infections, which reduces the efficacy of current options for outpatient treatment (Simmering et al. 2017) (Zilberberg and Shorr 2013). In this regard, fluoroquinolones are an important factor due to their high resistance rate in all age groups over 50 years old (Goldstein et al. 2019) (Ahmed et al. 2019). Overtreatment and incorrect use of antibiotics at the community (Bruxvoort et al. 2020) and hospital level (Palacios-Ceña et al. 2017) and deficiencies in the social care networks required for community support (Rosello et al. 2018) have also been indicated. One factor that is directly related to the increase in admissions is age (Simmering et al. 2017). In patients over 85 years old with multiple comorbidities, referral from long-term care facilities (Medina-Polo et al. 2015) delayed administration or delays in starting antibiotic treatment (Gharbi et al. 2019) (Álvarez Artero et al. 2019) also explain why the course of disease is worse in this age group.

The discussion is "poor". I think you can extend it for example by discussing about the economic impact of UTI, and the potential strategies for prevention of UTI.

Thank you for your comment. We have expanded the content of the discussion, adding the most striking results and information on the different sections of the study (pages 9-12). We have added information on costs to the introduction (page 3, paragraph 2), which reads as follows:

These hospitalizations have a high economic impact, primarily associated with the length of the hospital stay (Vallejo-Torres et al. 2018).

A final sentence, including the clinical significance of your findings is missing.

We have included a sentence at the end of the manuscript to emphasize the implications of this research. Page 10, paragraphs 6 and 7 reads as follows:

The results obtained in this study provide an accurate assessment of the changes in the national incidence of hospitalizations due to UTIs over 16 years and allow us to estimate the real burden of UTIs in older patients and their impact on our health system.

Further research is needed on the incidence of UTIs at the community level and on patterns of antibiotic use, which would help to improve the control of resistance and optimize clinical management, establishing clinical pathways for managing this type of infection in older patients. We also need to understand the factors associated with hospitalization in higher-risk patients, such as those with comorbidities or catheters, so we can establish preventive strategies that would reduce admissions and have a favorable impact on patient health and on the health system.

OTHER

Avoid the use of the word "elderly".

Thank you very much. We have replaced “elderly” with “older adults” in the text.

CONCLUSION

In conclusion, in my opinion the work, although not fully original, it seems to be of some interest and could further expand the knowledge of these infections in the elderly patients, but it has to be revised. Provided that these points are expanded, I would suggest it for publication in the Journal.

Thank you for your evaluation of our work.

Yours faithfully,

Ricardo Rodriguez

---

## [Decision Letter · Decision Letter 1]

24 Aug 2021

PONE-D-21-01514R1

TRENDS IN URINARY TRACT INFECTION HOSPITALIZATION IN OLDER ADULTS IN SPAIN FROM 2000-2015

PLOS ONE

Dear Dr. Rodríguez-Barrientos,

Thank you for submitting your manuscript to PLOS ONE. After careful consideration, we feel that it has merit but does not fully meet PLOS ONE’s publication criteria as it currently stands. Therefore, we invite you to submit a revised version of the manuscript that addresses the points raised during the review process.

ACADEMIC EDITOR:

There are sentences in Spanish in the text that require translation to English.

**"Análisis de tendencias (joint-point) global, por tipo de itu, por sexo y por grupo de edad"**

**"Por tipo de ITU"**

We look forward to receiving your revised manuscript.

Kind regards,

Justyna Gołębiewska

Academic Editor

PLOS ONE

Journal Requirements:

Reviewers' comments:

Reviewer's Responses to Questions

**Comments to the Author**

1. If the authors have adequately addressed your comments raised in a previous round of review and you feel that this manuscript is now acceptable for publication, you may indicate that here to bypass the “Comments to the Author” section, enter your conflict of interest statement in the “Confidential to Editor” section, and submit your "Accept" recommendation.

Reviewer #2: All comments have been addressed

2. Is the manuscript technically sound, and do the data support the conclusions?

Reviewer #2: Yes

3. Has the statistical analysis been performed appropriately and rigorously? 

Reviewer #2: Yes

4. Have the authors made all data underlying the findings in their manuscript fully available?

Reviewer #2: Yes

5. Is the manuscript presented in an intelligible fashion and written in standard English?

Reviewer #2: Yes

6. Review Comments to the Author

Reviewer #2: In this revision, the research carried out by the authors seems to have been deeply revised and expanded , according to the reviewer's suggestions. In concusion, taking note of the relevant changes made to the work, I would definitely suggest its publication in the Journal.

7. PLOS authors have the option to publish the peer review history of their article (what does this mean?). If published, this will include your full peer review and any attached files.

Reviewer #2: **Yes: **Angioni Davide

---

## [Author Response · Author response to Decision Letter 1]

27 Aug 2021

Dear Editor,

Sorry for the mistake, the two sentences have already been translated.

Kind regards

---

## [Editor Report · Decision Letter 2]

2 Sep 2021

PONE-D-21-01514R2

TRENDS IN URINARY TRACT INFECTION HOSPITALIZATION IN OLDER ADULTS IN SPAIN FROM 2000-2015

PLOS ONE

Dear Dr. Rodríguez-Barrientos,

Thank you for submitting your manuscript to PLOS ONE. After careful consideration, we feel that it has merit but does not fully meet PLOS ONE’s publication criteria as it currently stands. Therefore, we invite you to submit a revised version of the manuscript that addresses the points raised during the review process.

ACADEMIC EDITOR: Please submit a corrected version with the latest corrections indicated.

You submitted Revision 1 with the answers to previous reviews.

We look forward to receiving your revised manuscript.

Kind regards,

Justyna Gołębiewska

Academic Editor

PLOS ONE
---

## [Author Response · Author response to Decision Letter 2]

3 Sep 2021

We have replaced "Análisis de tendencias (joint-point) global, por tipo de itu, por sexo y por grupo de edad" and "Por tipo de ITU" with “Analysis of trends (joinpoint) globally and by UTI type, sex, and age group ”and“ By UTI type. Sorry for the mistake. We have submitted the updated 'Response to Reviewers', 'Revised Manuscript with Track Changes' and 'Manuscript' files.

Yours faithfully,

Ricardo Rodriguez

---

## [Editor Report · Decision Letter 3]

6 Sep 2021

TRENDS IN URINARY TRACT INFECTION HOSPITALIZATION IN OLDER ADULTS IN SPAIN FROM 2000-2015

PONE-D-21-01514R3

Dear Dr. Rodríguez-Barrientos,

We’re pleased to inform you that your manuscript has been judged scientifically suitable for publication and will be formally accepted for publication once it meets all outstanding technical requirements.

Kind regards,

Justyna Gołębiewska

Academic Editor

PLOS ONE
---

## [Editor Report · Acceptance letter]

15 Sep 2021

PONE-D-21-01514R3 

Trends in Urinary Tract Infection hospitalization in older adults in Spain from 2000-2015 

Dear Dr. Rodríguez-Barrientos:

I'm pleased to inform you that your manuscript has been deemed suitable for publication in PLOS ONE. Congratulations! Your manuscript is now with our production department. 

Kind regards, 

on behalf of

Dr. Justyna Gołębiewska 

Academic Editor

PLOS ONE